RRE-deleting self-inactivating and self-activating HIV-1 vectors for improved safety

Srinivasakumar Narasimhachar skumar15@slu.edu
Division of Hematology/Oncology, Department of Internal Medicine, Saint Louis University , Saint Louis, Missouri , USA
Decaro Nicola
Electronic publication date: 2013 Jun 4
Publication date: 2013
Volume: 1
Electronic Location ID: e84
Received 2013 Mar 27; Accepted 2013 May 17
Copyright: © 2013 Srinivasakumar
Copyright year: 2013
Copyright holder: Srinivasakumar
License: This is an open access article distributed under the terms of the Creative Commons Attribution License, which permits unrestricted use, distribution, and reproduction in any medium, provided the original author and source are credited.
License URL: https://creativecommons.org/licenses/by/3.0/

Keywords: RRE, Gene therapy, Vector, Lentivirus, HIV, HIV-1 vector, Self-inactivating, Self-activating

Funding: National Institutes of Health DK 53929 and AI 054211 The work in this study was supported by funds from the National Institutes of Health (DK 53929 and AI 054211). The funders had no role in study design, data collection and analysis, decision to publish, or preparation of the manuscript.

==============================
Retroviruses have been shown to efficiently delete sequences between repeats as a consequence of the template switching ability of the viral reverse transcriptase. To evaluate this approach for deriving safety-modified lentiviral vectors, we created HIV-1 vectors engineered to delete the Rev-response element (RRE) during reverse-transcription by sandwiching the RRE between two non-functional hygromycin phosphotransferase sequences. Deletion of the RRE during reverse-transcription lead to the reconstitution of a functional hygromycin phosphotransferase gene in the target cell. The efficiency of functional reconstitution, depending on vector configuration, was between 12% and 23%. Real-time quantitative PCR of genomic DNA of cells transduced with the RRE-deleting vectors that were selected using an independent drug resistance marker, which measured both functional and nonfunctional recombination events, indicated that the overall efficiency of RRE deletion of hygromycin phosphotransferase gene, was between 73.6% and 83.5%.

Introduction

HIV-1 is a complex retrovirus and as such codes for at least 9 proteins (Jeang et al., 1991). This is mainly achieved by the use of differentially spliced mRNAs. At least 30 distinct mRNAs have been identified in HIV-1 (Purcell & Martin, 1993). All the spliced forms originate from a single genome-length mRNA transcript. It therefore follows that coding regions for proteins can function as both introns and exons. The full-length mRNA is used for expression of Gag and Gag-Pro-Pol proteins. However the gag/pol coding sequence serves as an intron and is spliced out for expression of Env. Normally, intron-containing mRNAs are retained in the nucleus (Chang & Sharp, 1989). The virus therefore requires a mechanism for transporting incompletely spliced and unspliced mRNAs from the nucleus to the cytoplasm for either protein expression or for encapsidation of the full-length or genomic mRNA. In complex retroviruses, such as HIV-1 this is achieved by a regulatory protein, Rev. The Rev proteins binds to a structured RNA element present within the env coding region called the Rev-response element (RRE) (Hammarskjold et al., 1989; Malim et al., 1990). Rev protein then recruits host proteins such as Crm1 to effect nucleo-cytoplasmic transport of viral mRNAs (Fornerod et al., 1997).

Lentiviral gene delivery systems consist of packaging (or helper) plasmids that code for viral structural and regulatory proteins, and a gene transfer vector that contains the transgene expression cassette (Srinivasakumar, 2001). Expression of viral Gag/Gag-Pro-Pol proteins by the packaging construct requires appending the RRE sequence in the mRNA, and coexpression of the viral Rev protein (Smith et al., 1990).

Although HIV-1 based gene transfer vectors lack most of the viral coding sequences, it retains a small portion of the gag sequence, and also contains a 5′ splice donor site upstream of gag and frequently a 3′ splice acceptor site further downstream. Some vectors possess additional splice donor and splice acceptor sites. All gene transfer vectors also contain cis-sequences for encapsidation, reverse-transcription, and integration. HIV-1 based vectors also contain the RRE sequence. In addition to its role in nucleo-cytoplasmic transport, Rev and RRE may also have a role in encapsidation of the genomic RNA in virus particles (Brandt et al., 2007; Cockrell et al., 2011). The RRE, however, is usually dispensable in the target cell for expression from the transgene expression cassette (Srinivasakumar, 2001).

The retroviral reverse transcription process is complex and involves at least two obligatory template switches by the reverse-transcriptase (RT) (Delviks-Frankenberry et al., 2011). This ability of the RT to ‘jump’ is likely responsible for elimination of direct repeats, and sequences inserted between the repeats during reverse transcription. The sequence requirements for efficient deletion of direct repeats, as well as the inserted sequences, have been well documented (Delviks-Frankenberry et al., 2011; Onafuwa-Nuga & Telesnitsky, 2009).

Retroviruses, including lentiviruses, contain two copies of genomic RNA. Template switching by RT between these two molecules can result in recombinant viruses (Delviks-Frankenberry et al., 2011; Onafuwa-Nuga & Telesnitsky, 2009). When a vector transduced cell is coinfected with a wild-type HIV, it can mobilize the vector derived RNA to other untransduced cells (Evans & Garcia, 2000). One way to prevent mobilization is by creating self-inactivating (SIN) vectors. The traditional approach is by introducing a debilitating mutation in the U3 region of the 3′ LTR (Zufferey et al., 1998). As an alternative approach, we surmised that we could exploit the template-switching property of retroviral RTs, including that of HIV-1 RT, to design a gene transfer vector that would eliminate the RRE during the infection process. Elimination of the RRE during reverse transcription would result in reduced ability for the vector sequences to be mobilized from the transduced cells upon coinfection of the cell with wild-type HIV-1 (Lucke, Grunwald & Uberla, 2005). To this end, the RRE sequence in the vector was sandwiched between two non-functional hygromycin phosphotransferase (Hyg) sequences. The results indicated that the deletion of RRE during the reverse transcription process reconstituted a functional Hyg gene depending on the site of recombination within the Hyg inactivating mutation.

Materials and methods

Plasmids

The packaging plasmid pgp3virin (Srinivasakumar & Schuening, 1999) and the envelope expression plasmid, pMD.G encoding VSV-G (Naldini et al., 1996) have been described previously. The wild-type vector was derived form pTR167 (Rizvi & Panganiban, 1993) and was modified to contain a frame-shift mutation in gag and an inactivated nef open reading frame. It contains a simian virus 40-Hyg (SV-Hyg) cassette at the unique NheI site of the truncated env sequence upstream of the RRE (Fig. 1). The RI(-) vector and the SacII(-) vector were derived from wild-type vector, by digestion with EcoRI or SacII followed by repair using T4 polymerase and religation, respectively. The Aug(-) vector was created using PCR based mutagenesis as follows: The Hyg gene was amplified using a pair of primers but with the sense primer targeting the 5′ coding of the Hyg gene lacked the AUG codon. The amplified product was positioned downstream of the SV40 early promoter between the BamHI and XhoI restriction enzyme sites. The dual-Hyg containing vectors RI(-)/Aug(-) and SacII(-)/Aug(-) were created in multiple steps. First the Hyg sequence lacking the AUG codon was positioned downstream of the RRE and between BamHI and XhoI sites. Next the SV-Hyg cassette from RI(-) and SacII(-) vectors was released with NheI and ligated into the NheI site upstream of the RRE to give RI(-)/Aug(-) and SacII(-)/Aug(-), respectively (Fig. 1). The ΔRRE vector was created by digestion of SacII(-)/Aug(-) vector with EcoRI and dropping the EcoRI fragment between the EcoRI sites of the two Hyg sequences, thus eliminating the RRE, prior to religation (Fig. 1).

Figure 1 Schematic representation of HIV-1 provirus and RRE-deleting HIV-1 vectors.

(A) The genetic organization of HIV-1 proviral clone pNL4-3 depicting the 5′ and 3′ long-terminal repeats (LTRs), protein coding regions, and the Rev-response element (RRE). Important restriction enzyme sites used in the creation of the gene transfer vectors are shown on a horizontal line below. (B) Schematic of RRE-deleting gene transfer vectors. The wild-type vector is shown at the top in B. It contains a single Hyg sequence driven under control of simian virus 40 early promoter (SV) that is positioned upstream of the RRE. The RRE-deleting vectors contain two non-functional Hyg sequences, one upstream, and the other downstream of the RRE. There are two versions of this vector: The RI(-)/Aug(-) version contains a mutation at the EcoRI (RI) site, while the SacII(-)/Aug(-) version contains a mutation at the SacII site, in the upstream Hyg gene. Both vectors carry a deletion of the AUG initiator codon (Aug(-)) in the Hyg sequence downstream of the RRE. A vector that has the RRE sequence deleted, ΔRRE, is shown at the bottom.

Cells

Human embryonic kidney 293T (HEK293T) cellswere obtained from American type culture collection (ATCC; catalog number SD-3515). The HEK293T cells were maintained in Dulbecco’s modified Eagle’s medium supplemented with 2 mM L-glutamine, 100 U/ml penicillin, 100 µg/ml streptomycin and 10% heat-inactivated fetal bovine serum (Hyclone/ThermoFisherScientific, USA). HeLa cells were maintained in Iscoves medium supplemented with 2 mM L-glutamine, 100 U/ml penicillin, 100 µg/ml streptomycin and 10% heat-inactivated newborn calf serum (Hyclone/ThermoFisherScientific, USA).

Preparation of vector stocks

The packaging plasmid, pgp3virin encoding Gag/Gag-Pro-Pol and all accessory and regulatory HIV-1 protein (3.75 µg), VSV-G envelope expression construct, pMD.G (0.2 µg) and the gene transfer vector (7.5 µg) were transfected into 293T cells in T25 flasks using the CaPO4 method as previously described (Srinivasakumar, 2002). For some experiments, in place of pgp3virin, we used an alternative packaging and helper constructs consisting of pGP-HIV-1 350 RRE encoding Gag and Gag-Pro-Pol (1.5 µg), pCMVtat (0.1 µg) and pCI-Rev (0.1 µg) (Srinivasakumar, 2008) for transfection in 6-well plates. Vector stocks were harvested 72 h later, clarified by centrifugation at 2500 × rpm (1400 × g) at 4°C for 15 min and stored at −80°C in aliquots or used immediately for titration on HeLa cells.

HIV-1 p24 ELISA

HIV-1 p24 ELISA was done using a commercial kit obtained from PerkinElmer (Massachusetts, USA) using the recommended protocol.

Transduction of HeLa cells

HeLa cells were seeded into 6-well plates (200,000 cells/well) one day prior to infection with vector stocks. Next day, each well received 200 µl, 20 µl or 2 µl of vector stock in 1 ml of Iscoves growth medium containing 8 µg/ml polybrene. After overnight incubation at 37°C, the polybrene was diluted with 2 ml of complete growth medium. The following day, the medium was replaced with fresh medium containing either hygromycin B (Calbiochem, Darmstadt, Germany) (200 µg/ml) or G418 (Life Technologies, NY, 1 mg/ml). This was replaced every 3–4 days with fresh medium containing the selection agent. After 12–14 days, when no live cells were noted in control untransduced cultures, and colonies were visible in test cultures, the medium was aspirated, and the cells fixed and stained using 0.5% crystal violet in 50% methanol before enumeration.

Isolation of genomic DNA from transduced cells

This was done using Qiagen DNeasy miniprep kits (Qiagen, Maryland, USA) according to the manufacturer’s recommended protocol and included an Rnase I treatment step.

Real-time or quantitative PCR (qPCR)

This was done in a Bio-Rad CFX96 thermocycler using iQ SYBR green supermix in 20 µl reaction volumes using 60 ng of template DNA and 200 nM concentration of each primer. A two-step PCR was used with an annealing and extension temperature of 63.1°C and a denaturation temperature of 95°C for a total of 40 cycles. The SYBR Green fluorescence was detected during the annealing and extension step (63.1°C). A final melt-curve analysis was done by ramping up the temperature from 65°C to 80°C in 0.5°C increments. Ten-fold dilutions of genomic DNA transduced with control ‘wild-type’ vector were used to generate standard curves. In addition to using primer pairs targeting RRE and gag, we also tested each genomic DNA sample using primers against β-actin to ensure that equal amounts of DNA samples (as deduced from spectroscopically estimated DNA concentrations) could be amplified to a similar extent, and that there were no PCR inhibitors in the final reaction mix. The primer sequences used to amplify gag, RRE or β-actin are shown in Table 2. Relative quantities of RRE to gag were estimated as previously described (Pfaffl, 2001) and detailed in the footnote to Table 3.

Results

RRE-deleting vectors

We created HIV-1 derived vectors designed to eliminate the RRE during reverse transcription (Fig. 1). One Hyg sequence, under control of the simian virus 40 (SV40) immediate early promoter, was placed upstream of the RRE at the unique NheI site in the vector backbone. The upstream Hyg sequence contained a frame-shift mutation at either the EcoRI site or at the SacII restriction enzyme sites. The second Hyg sequence was positioned downstream of the RRE, between the BamHI site in the second coding exon of Rev and the XhoI site in Nef. The downstream Hyg sequence contained a deletion of the initiator AUG codon (Aug(-)). Control vectors encoded a single copy of either the frame-shifted (RI(-) or SacII(-)) or the AUG(-) versions of the Hyg gene under control of the SV40 early promoter. The ‘wild-type’ control vector had an intact Hyg gene under control of the SV40 early promoter. A vector that recapitulated the RRE-deleted form (ΔRRE) was also created. This vector lacked a 3′ splice site.

A vector lacking the RRE is transduced less efficiently than one with RRE

The Rev-RRE transport pathway is crucial for transport and expression of proteins from full-length HIV-1 (Gag/Gag-Pro-Pol) or HIV-1 mRNAs containing introns (e.g., Env). Since HIV-1 based gene delivery vectors have most of their coding regions removed, it is not clear if Rev and RRE are essential for packaging and transduction of such vectors. To this end, we compared efficiency of gene delivery by the wild-type vector containing an RRE with a vector lacking RRE (ΔRRE) (Fig. 1). The wild-type and ΔRRE vectors were packaged in 293 T cells, as previously described, and the resultant titers (not normalized to p24 levels) were determined on HeLa cells following selection with hygromycin B. The results of this experiment are shown in Fig. 2. The wild-type vector titer (1.3 ± 0.1 × 104 cfu/ml) was approximately 61-fold higher than that of the ΔRRE vector (2.2 ± 0.1 × 102 cfu/ml) that exhibited colonies only at the highest volumes tested. These results indicate that Rev and RRE are essential for efficient packaging and transduction of gene delivery vectors based on HIV-1 despite lacking most of the protein coding regions. This observation is supportive of our efforts to create an RRE-deleting HIV-1 vector for gene delivery.

Figure 2 A vector lacking RRE is severely crippled for transduction into HeLa cells.

Vector stocks were produced by transient transfection of 293T cells with the packaging plasmid pGP-HIV-1 350 RRE, plasmids encoding Rev (pCI-Rev) and Tat (pCMVtat), a VSV-G envelope expression construct, and a gene-transfer vector (wild-type or ΔRRE), as described in Materials and Methods. HeLa cells were transduced with the indicated vectors (200 µl or 20 µl) or vector-free supernatant (mock) and resultant colonies were fixed and stained as described in Materials and Methods. The transfections were done in parallel (biological replicates) on the same day. The calculated titers in colony forming units (CFU)/ml is shown on the right.

Functional reconstitution of hygromycin resistance

Vector stocks for each of the RRE-deleting gene transfer vectors (Fig. 1), as well as control vectors were produced in 293T cells as previously described. Virus containing medium was harvested 72 h post-transfection, clarified by centrifugation, and used for the infection of HeLa cells. The transduced cells were subjected to selection using hygromycin B and the resultant colonies were fixed, stained, and enumerated two weeks later.

Hygromycin-resistant colonies obtained from test and control vectors from two such experiments are shown in Fig. 3. The control wild-type vector encoding a functional Hyg gene had the highest density of colonies followed by the SacII(-)/Aug(-) and the RI(-)/Aug(-)vectors. In contrast, the vectors encoding the individual mutated Hyg genes resulted in no hygromycin-resistant colonies (data not shown). Reconstitution of a functional hygromycin resistance gene was readily apparent for the RI(-)/Aug(-) and SacII(-)/Aug(-)vectors.

Figure 3 Reconstitution of functional Hyg gene during transduction of HeLa cells with RRE-deleting HIV-1 vectors.

Vector stocks were produced by transient transfection of 293T cells with a packaging plasmid (pgp3virin), a VSV-G envelope expression construct and the indicated vector, as described in Materials and Methods. Transfections were done in duplicate. Each individual vector stock was used for infection of HeLa cells in 6-well plates using the indicated amounts (200 µl, 20 µl or 2 µl). The cells were then subjected to selection using hygromycin B as described in in the text for 12–14 days. The resultant colonies were fixed and stained with crystal violet in 50% methanol and enumerated. The results of two independent experiments (Expt 1 and Expt 2) are shown. Each experiment consisted of two parallel transfections (biological replicates) done on the same day. The calculated titers in colony forming units (CFU)/ml as well as the p24 levels in ng/ml are shown to the right of each titration.

To more accurately determine the efficacy of reconstitution, the vector titers determined from Fig. 2 were normalized using p24 levels in the vector-containing supernatants. These results are summarized in Table 1. The efficiency of functional reconstitution, determined from two independent experiments, was 23.3 ± 2.18% for the SacII(-)/Aug(-) vector and 11.93 ± 1.94% for the RI(-)/Aug(-) vector. This difference between SacII(-)/Aug(-) and RI(-)/Aug(-) vectors was statistically significant (Student’s t-test, p < 0.05).

Table 1 Functional reconstitution of Hyg gene in RRE-deleting HIV-1 vectors.

	Expt 1	Expt 2	Normalizedb	
	CFU/p24a
(mean ± SD)	CFU/p24a
(mean ± SD)	CFU/p24
(mean ± SD)	
Wild-type	5008 ± 668	1206 ± 52	100 ± 0.00	
RI(-)/Aug(-)	666 ± 209	127 ± 96	11.93 ± 1.94	
SacII(-)/Aug(-)	1090 ± 80	300 ± 16	23.32 ± 2.18	
Notes.

a The p24 levels (in ng/ml) in vector stocks were determined using a commercial ELISA kit (Perkin-Elmer, Boston, MA).

b The colony forming units (CFU)/p24 data in each experiment was normalized to that of wild-type vector which was set at 100%. The mean and standard deviation (SD) of combined data from both experiments are shown in the last column.

Table 2 Table showing primer sequences.

Primer	Laboratory designation	Sequence (5′–3′)	Parent GenBank Accession No.	
gag sense	SK97	GAACGATTCGCAGTTAATCC	M19921	
gag antisense	SK98	GATGCACACAATAGAGGACTGC	M19921	
RRE sense	SK140	ATCAAACAGCTCCAGGCAAG	M19921	
RRE antisense	SK141	ACAGCAGTGGTGCAAATGAG	M19921	
β-Actin sense	SK108	AGAAAATCTGGCACCACACC	NM_001101	
β-Actin antisense	SK109	AGAGGCGTACAGGGATAGCA	NM_001101	

Table 3 Real-time qPCR analysis of genomic DNA of vector-transduced cells to determine recombination frequency.

Vector	CqGag
(Mean ± SD)	CqRRE
(Mean ± SD)	Fold reduction
in RREa	RRE deletion
frequencyb	
Wild-type	23.46 ± 0.25	23.93 ± 0.44	1.00	0%	
RI(-)/Aug(-)	23.40 ± 0.15	26.59 ± 0.50	6.25	83.5%	
SacII(-)/Aug(-)	23.20 ± 0.06	25.67 ± 1.40	3.85	73.6%	
ΔRRE	26.80 ± 2.44	37.55 ± 0.64	NAc	100%	
Notes.

a Fold reduction of RRE with reference to Wild-type vector was calculated using the formula: ErΔCq(r)÷EtΔCq(t), where Er = efficiency of amplification of reference (gag) sequence; Et = efficiency of amplification of target (RRE) sequence; ΔC q(r) = Quantification cycle (C q) difference between control (wild-type) and test vectors for reference sequence (gag); ΔC q(t) = C q difference between control (wild-type) and test vectors for target sequence (RRE). i.e., 2.00gagΔCq(gag)(23.46- vector)÷1.94RREΔCq(RRE)(23.93- vector). Where 1.94 and 2.00 refer to PCR efficiency of RRE and gag primer pairs determined from the slope of standard (dose-response) curves.

b Normalized to wild-type vector as follows: (1-(1/fold reduction in RRE)) × 100.

c NA = Not applicable as the fold-reduction cannot be calculated using the above formula for RRE value of zero.

Real-time quantitative PCR (qPCR) can be used to determine recombination frequency

Selection with hygromycin B ensures detection of only functional reconstitution of the gene and overlooks recombination events that might have resulted in elimination of RRE but did not result in reconstituting a functional Hyg gene. To determine frequency of all RRE-deleting events, including those events not resulting in reconstitution of a functional hygromycin gene, we introduced a neomycin resistance gene (Neo) into the RI(-)/Aug(-) and SacII(-)/Aug(-) vectors downstream of the 3′ Hyg sequence (Fig. 1). The Neo gene was translated by virtue of an encephalomyocarditis virus derived internal ribosome entry site (IRES) situated at its 5′ end. The vectors were used for transduction of HeLa cells that were then selected using G418. Pools of G418-resistant colonies (at least 50 colonies for each vector) were prepared and genomic DNA isolated using Qiagen DNeasy kits for each of the two RRE-deleting vectors as well as the control ΔRRE vector. Genomic DNA from cells transd uced with the ‘wild-type’ vector harboring an intact RRE and a functional Hyg gene were also obtained.

We used qPCR to detect the efficiency of recombination or elimination of the RRE sequence during the reverse-transcription process. To this end we designed PCR primers targeting the RRE and gag sequences (Table 2). The optimal temperature for each primer pair was first determined using a temperature gradient. Based on these preliminary studies, we were able to design a qPCR assay that could reliably amplify both targets (RRE and gag, but amplifed independently in separate wells) during the same PCR run with high efficiency and specificity. Using these optimized conditions we determined the quantitation cycle (Cq) values for RRE (target) and gag (reference) sequence using genomic DNA of target cells transduced with each of the vectors (Table 3). The gag sequence was used as the reference sequence since it was present in the remnant gag sequence of all the vectors used in this study. The fold-difference in RRE between the vectors was determined by normalizing the results to the wild-type vector. The relative RRE values for wild-type, RI(-)/Aug(-) and SacII(-)/Aug(-) vectors were 1.0, 0.16 and 0.26, respectively. This corresponded to an efficiency of deletion of RRE of 83.5% and 73.6% for the RI(-)/Aug(-) and SacII(-)/Aug(-) vectors, respectively. The difference between the RI(-)/Aug(-) and SacII(-)/Aug(-) vectors was not statistically significant (Student’s t-test, p > 0.05). Genomic DNA of cells transduced with the ΔRRE vector exhibited gag Cq values that were 3.3 cycles higher than the other test vectors indicating approximately 10-fold reduction in quantity. This can be explained by the inherently low titers of ΔRRE vector stocks resulting in lower multiplicities of infection (MOI) than for the other test vectors exhibiting higher titers.

Discussion

In the present study, we designed RRE-deleting HIV-1 vectors by sandwiching a 1.2 kb env sequence containing the RRE between two Hyg sequences. The upstream Hyg sequence had a mutation at either the EcoRI or the SacII restriction enzyme site. The downstream Hyg lacked the AUG initator codon. All three of these mutations, when tested individually in vectors, were found to render the hygromycin non-functional. The RRE-deleting vectors showed functional reconstitution of the Hyg gene, with the RI(-)/Aug(-) vector showing a lower efficiency than the SacII(-)/Aug(-) vector. This result can be explained as follows: The EcoRI site is at position 243 while the SacII site is at positon 777 of the 1053 nt long Hyg gene (Fig. 4A). The distance between the upstream and downstream RI (or SacII) sites in the Hyg sequences sandwiching the RRE is 2.3 kb (Fig. 4B). If we assume equal rates of recombination events along the Hyg gene, then functional reconstitution for the SacII(-)/Aug(-) vector should be about 2.9-fold greater than for the RI(-)/Aug(-) vector. The experimentally determined rate of functional reconstitution for the SacII(-)/Aug(-) vector was 2.00 ± 0.51-fold higher than that of the the RI(-)/Aug(-) vector (Table 1) and approximates the theoretical calculation.

Figure 4 Schematic showing location of restriction enzyme sites in the context of Hyg sequence and the RRE-deleting vectors.

(A) The location of the EcoRI (RI) and SacII restriction enzyme sites within the Hyg sequence. (B) The distance between the restriction enzyme sites (RI or SacII) in the context of the RRE-deleting vectors (2.3 kb) is indicated, as is the size of the HIV-1 env sequence (1.2 kb) containing the RRE and situated between the two Hyg sequences.

An alternative explanation for functional reconstitution is recombination at the DNA level in the producer cell. A functional reconstitution would require the elimination of the RRE/env sequence. Consequently, the resulting vector derived mRNA from the recombinant plasmid would not be efficiently packaged and transduced (Fig. 2). Another possibility is aberrant splicing from an upstream exon to a site downstream of the AUG mutation in the 3′ Hyg sequence. Again any splicing occurring upstream of the RRE would eliminate the RRE and inactivate the vector. The high efficiency in the deletion of the RRE sequence following transduction (see discussion of qPCR below), makes this also unlikely.

We also looked at the overall efficiency of deletion of the intervening RRE sequence, in the absence of selection for Hyg, using an independent Neo selection marker. This measured both functional and nonfunctional reconstitution of hyg sequence. The genomic DNA of transduced cells selected with Neo was analyzed using qPCR to estimate RRE and gag copy numbers and the efficiency of deletion of RRE was determined. The results of these experiments (Table 3) showed that the RRE was deleted at an efficiency of 73.6% and 83.5% for the RI(-)/AUG(-) and SacII(-)/Aug(-) vectors, respectively. These differences in efficiency of deletion of RRE were not statistically significant. Similar rates of RRE deletion were expected for both vectors, as they both contained the same repeat sequences (with the exception of the mutated restriction sites) and the same interjacent sequences.

The requirements for recombination during infection of target cells by retroviruses, including lentiviruses, have been extensively characterized by several groups (reviewed in Delviks-Frankenberry et al., 2011; Onafuwa-Nuga & Telesnitsky, 2009). Direct repeat sequences are deleted quite efficiently. This deletion appears to be proportional to the length of the direct repeats. Thus, the efficiency of direct repeat deletion for lengths of 117 bp, 284 bp and 971 bp was found to be 5%, 27% and 60%, respectively, for Moloney murine leukemia virus, while for the same lengths, it was found to be 6%, 19% and 81%, respectively, for HIV-1 (An & Telesnitsky, 2001). The efficiency of deletion could be enhanced by increasing the distance between the repeats. Delviks and coworkers created Moloney murine leukemia virus based vectors containing 701-bp repeats. The repeats were separated by intervening sequences ranging from 100 bp to 3.5 kb. The efficiency of deletion of repeats for vectors with an intervening sequence of 1.5 kb was over 90% (Delviks & Pathak, 1999). For HIV, the recombination rate was 42.4%, 50.4%, and 47.4% for markers separated by distances of 1.0 kb, 1.3 kb, and 1.9 kb (Rhodes, Wargo & Hu, 2003). In the last study, the recombination measured intermolecular recombination between the two RNA genomes copackaged into the virion.

In general, lentiviruses therefore appear to be more prone to undergo recombination events than gammaretroviruses (Delviks-Frankenberry et al., 2011; Onafuwa-Nuga & Telesnitsky, 2009). This difference cannot be attributed to differences in the reverse transcriptases among the viruses, such as template switching rates that can be influenced by RNAse H activity, or processivity (Hwang, Svarovskaia & Pathak, 2001). Rather, the evidence suggests that this is likely to be due to the higher efficiency of copackaging of RNA molecules into budding lentiviruses. The copackaging of RNA molecules is directed by the dimer initiating site (DIS). Thus, the palindromic GCGCGC sequence allows copackaging of two distinct parent viruses in accordance with the Hardy-Weinberg Equilibrium to an efficiency approaching 50%. When the DIS was complementary in the two parent viruses, the recombination efficiency was further increased and approached 70% (Moore et al., 2007).

Other investigators have used the template switching ability of reverse transcriptase to design safety modified gene delivery vectors. One such example is the use of this ability to delete the packaging signal present in the 5’ untranslated region of retroviruses (Delviks, Hu & Pathak, 1997; Julias, Hash & Pathak, 1995). The RRE-deleting HIV-1 vector described here is, likewise, a safety modified lentivirus vector. A vector lacking RRE is less likely to be mobilized from the target cell upon infection with a replicating wild-type virus because a vector RNA that is trapped in the nucleus is not available for encapsidation (Helga-Maria, Hammarskjold & Rekosh, 1999). Rev and RRE have also been implicated to play a role in encapsidation of genomic RNA (Brandt et al., 2007; Cockrell et al., 2011) and this can also decrease mobilization. In contradistinction to this observation, we have reported high-titered Rev-free HIV-1 based gene delivery systems (Srinivasakumar, 2011). This would suggest that the effect of Rev and RRE on encapsidation may depend on the precise configuration of the gene-transfer vector. Nevertheless, a combination of the traditional self-inactivating mutation in the U3 region of the 3′ LTR together with a RRE-deleting vector configuration would considerably enhance the safety of HIV-1 based lentivirus vectors for gene therapy applications.

I thank Dr. Michail Zaboikin for critical review of the manuscript.

Additional Information and Declarations

Competing Interests

Author Contributions

I declare that I have no competing interests.

Narasimhachar Srinivasakumar conceived and designed the experiments, performed the experiments, analyzed the data, contributed reagents/materials/analysis tools, wrote the paper.

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
