# Peer review of "RRE-deleting self-inactivating and self-activating HIV-1 vectors for improved safety"

_PeerJ, doi:10.7717/peerj.84_

## Round 0.1 · original submission · Major Revisions

Dear corresponding authors, I believe your article can be published after addressing the reviewers' comments.

·

Basic reporting

The article entitled "RRE-deleting self-activating HIV-1 vector" is well written and structured; the author designed and created gene transfer vectors (lentivirus vectors) to eliminate the RRE (during reverse transcription). However the article needs some minor revisions.

- In the Introduction section (line 52) the authors write and comment their results. The Introduction should not include the results and comment to the results, the authors should explain the purpose of the work and what they did. So the sentence “The results indicated thet the deletion........” at the end of the Introduction does not seem very appropriate.

Experimental design

-In material and methods section, in the Plasmids section, the authors write about the packaging plasmid pcgp3 and that they have been described previously. The authos should insert a sentence and indicate the proteins that are expressed or what this plasmid express.

-The authos should write a sentence, in the Material and method section- in the plasmids section, about the frameshift mutation in the upstream Hyg sequence, and how it was obtained.

- The author, if possible, should also improve the picture, to better describe the steps to obtain the various vectors (from the wild type vector).

- In material and methods section, in the real time or quantitative pcr (qPCR) section the authos should control the simbol of the concentration of each primer, nm means nanomolar ? Is it correctly written? Perhaps, it is better to be write nM.

- In Figure 2, the author should indicate 200 μl and 20 μl to the base of the figure.


- Result section : Line 156 “Selection with hygromycin B ensures detection of only functional reconstitution of the gene and overlooks recombination events that might have resulted in elimination of RRE but did not result in reconstituting a functional Hyg gene.” “To determine frequency of all RRE-deleting events,……… ”

The authors should better explain this sentence.

Validity of the findings

No Comments

Additional comments

- The authors should perform the revisions indicated.

-In the discussion section the authors write “The RRE-deleting HIV-1 vector described here is, likewise, a safety modified lentivirus vector.”

Are there possible application of this vectors for therapeutic purpose?
If so, the authors should discuss this topic in the discussion section.

Reviewer 2 ·

Basic reporting

Frequent deletion of direct repeats is observed in the genome of HIV-1 due to recombination events during reverse transcription. In a very interesting study Srinivasakumar utilized this feature to delete the RRE sequence after a single round of infection in an HIV-1 derived lentiviral vector. This was possible by surrounding the RRE with two defective Hygromycin resistance gene sequences carrying different inactivating mutations. After transduction the author demonstrated the functional reconstitution of the resistance gene, most likely due to RT-mediated recombination. The RRE deletion frequency was furthermore quantified by qPCR analyses. The detected recombination frequencies in both assays are consistent with previously published reports. To my knowledge this is the first attempt to target the Rev/RRE system by RT-mediated recombination. The ability of RRE deletion by recombination could be an interesting tool for the design of lentiviral vectors. The manuscript is structured in an appropriate way and is written in a comprehensible fashion. The experiments, results and conclusions are presented clearly. However, additional control experiments are needed to support some of the conclusions drawn by the author. Especially a few major points in “Basic Reporting” and “Experimental Design” have to be addressed before I can recommend the paper for publication in PeerJ.

Major points
1) The importance of co-packaging of two genomic HIV-1 RNAs for recombination is discussed (Lines 219 to 229). However, co-packaging was not explained before. In addition, co-packaging of two (different) genomic RNA molecules is also the basis for vector mobilization after co-infection of transduced cells with HIV-1. Therefore, encapsidation of two genomic RNAs into HIV-1 particles and its role in recombination and mobilization should already be explained in the introduction.
2) The HIV-1 Rev protein does not only mediate the nuclear export of the genomic RNA but nuclear export mediated by Rev/RRE is also important for an efficient encapsidation process (see Cockrell et al. 2011 doi:10.1186/1742-4690-8-51 and Brandt et al. 2007 doi:10.1371/journal.ppat.0030054). This explains why “Rev and RRE are essential for packaging and transduction” (see Line 127 and 128) and can also explain the strongly reduced titer of the DeltaRRE vector. Therefore, the influence of Rev/RRE on encapsidation should be mentioned/discussed in the manuscript.
3) Importantly, deletion of the RRE in an HIV-1 derived vector was previously shown to efficiently reduce mobilization (Lucke et al. 2005 doi:10.1128/JVI.79.14.9359–9362.2005). This strengthens the rational behind the approach described and should be mentioned.
4) Results: Line 131 to 134: Mention that the titer is not normalized to p24 levels in this experiment. This is okay when the same packaging plasmids were used for the wild-type and the DeltaRRE vector production in parallel transfection experiments. Please state the packaging plasmids used in the text or the figure legend. Please comment whether transfections were done in parallel at the same day.
6) Results or Materials and Methods: Please mention which packaging plasmids were used in the experiments shown in Figure3 and Table3.
7) Results: Quantification of gag and RRE sequences in Table3 shows a strong reduction of gag sequences after transduction of the DeltaRRE vector. This perfectly fits to the reduced titer (Lines132-133). This should be mentioned in the manuscript. However, a certain amount of RRE sequences could also be detected although the vector does not contain the RRE. Please provide possible explanations for this finding. Could it be due to recombination between co-packaged helper RNA (gagpol-RRE mRNA) and vector RNA followed by RT and integration? Please speculate on that.

Minor points
Title:
self-inactivating HIV-1 vectors
Abstract:
first sentence: template switching ability of the viral reverse transcriptase
last sentence: of cells transduced with the RRE-deleting vectors which were selected using an
Entire Text:
-please limit the use of the word “message” and use RNA/mRNA/transcript instead
-wild-type not Wild-type in the text (e.g. Line 57)
Line 15, 2nd sentence: This is mainly achieved by the use of
Line 17, 3rd sentence: please add citation
Line 21, 8th sentence: please add citation
Line 23: delete second “or”: expression or for encapsidation of the full-length genomic RNA.
Line 29: Lentiviral gene delivery
Line 33: Add citation to the last sentence of the paragraph
Lines 34-40: Add a citation for this paragraph
Line 50: upon coinfection of the cell with wild-type HIV-1.
Line 54: in relation to the Hyg inactivating mutation.
Line 56: plasmid pcgp3virin encoding Gag/GagPol and all accessory and regulatory HIV-1 proteins
Line 56: and the expression plasmid, pMD.G encoding VSV-G
Line 56: pMD.G is not mentioned in the citation Srinivasakumar, Schuening 1999 please add an appropriate citation
Line 60: add link to Figure1: upstream of the RRE (Figure1).
Line 69: delete pN-FS-
Line 71: add link to Figure1: respectively (Figure1).
Line 73: add link to Figure1: prior to religation (Figure1).
Line 74: mention once that these are HEK293T cells to describe them unambiguously
Line 85: The plasmids are called “pGP-HIV-1 350 RRE” pCI-HIV-Rev” in the mentioned citation. Indicate them unambiguously.
Line 85: indicate the amount of transfected DNA (pGP-HIV-1 350 RRE, pCMVtat and pCI-HIV-Rev)
Line 101: nM not nm
Line 102: Please mention the temperature at which you detected the SYBRGreen fluorescence.
Line 113: We created HIV-1 derived vectors
Line 124 to 136: Mention that the DeltaRRE vector does not contain a 3’ splice site.
Line 136: … protein coding regions. This fact is supportive
Line 171: “both targets during the same PCR.” Rephrase the sentence to clarify that you did not use a multiplex PCR but two separate PCRs with the same cycling conditions. (In case you use a SYBRGreen-based multiplex PCR setting please add a detailed description to the Material and Methods section and show appropriate controls demonstrating the performance of the assay.)
Line 232: 5’ untranslated region of retroviral vectors
Line 235: efficiency
Figure1 legend for B: The wild-type vector is shown at the top in B. It contains a … simian virus 40 early promoter (SV) which is positioned …
Figure1: This figure is very valuable for the paper. It clearly depicts the cloning strategy and the composition of the vectors. Please also add lines between the BamHI+XhoI sites in the provirus and the downstream Hyg sequence in the RI(-)/Aug(-) vector. Additionally, please add lines between the EcoRI sites in SacII(-)/Aug(-) and the DeltaRRE vector, too.
Figure2 legend: Indicate whether you infected with two different supernatants harvested after two independent transfection experiments or if you infected two cultures with the same supernatant.
Figure2: Please add the calculated titers of each single infection to the figure (e.g. one value for Infection1 and one value for Infection2 at the right side of the wild-type image.)
Figure2: Add 200 µl and 20 µl at the bottom of the figure as in Figure3.
Figure3: Add the calculated titers for each infection to the figure (e.g. at the right side of the images). Is the label Expt1 and Expt2 consistent with Table1 for all plates?
Table1: Does Expt1/Expt2 refer to the same experiments as shown in Figure3?
Table3: When a “Delta” is used the letters overlap. A fold reduction of 0.16 would actually mean an increase. Please give the reciprocal values 1/0.16 = 6-fold reduction. The legend then should read “Normalized to wild-type vector as follows: (1-1/fold reduction in RRE)x100”

Experimental design

Major points
1) You did not filtrate the infectious supernatants of the producer cells through a filter (e.g. 0.45 µm). Please provide data that you do not (inadvertently) transfer Hyg- or Neo-resistant producer cells that could be counted as transduced colonies after the selection procedure.
1) The inactivating mutations in the Hyg sequences are clearly indicated in the manuscript. Furthermore, the single mutated genes in control vectors prevented the production of Hyg-resistant colonies. In contrast, the vectors RI(-)/Aug(-) and SacII(-)/Aug(-) produce Hyg-resistant colonies after transduction. The most likely explanation is indeed the reconstitution by recombination during the RT reaction. However, an alternative explanation could be that these vectors are able to produce functional Hygromycin phosphotransferase without recombination and before RT. This could for example be possible due to aberrant splicing events/usage of an alternative start codon/reinitiation after an upstream ORF. It is therefore necessary to prove that the vectors RI(-)/Aug(-) and SacII(-)/Aug(-) do not express a functional Hyg-resistance gene before RT. (Possible control experiment: Stable transfections of all vectors followed by Hyg-selection).
2) In Delviks et al. 1997 J Virol (cited in the manuscript) Neo-selection after transfection partially induced DNA rearrangements in a retroviral vector that resembled those changes observed after RT-mediated recombination (see Fig. 2A in this publication). Similar to the vector used in this publication you introduced the IRES-Neo sequence downstream of the Hyg(inactive)-RRE-Hyg(inactive) sequence. After transduction and Neo-selection you analyzed the RRE deletion frequency by qPCR. Please provide explanations/data demonstrating that such RT-independent rearrangements do not manipulate your results.
3) Materials and Methods:
Line 96: Please describe how the Neo selection with G418 was done. Furthermore, describe the p24 ELISA.

Validity of the findings

Minor Points
1) Lines192-194: Recombination events often occur during minus strand synthesis (Delviks-Frankenberry et al. 2011 doi:10.3390/v3091650). After the minus strand transfer the RT proceeds from the 3’ to the 5’ end of the viral RNA. Therefore, the rate of functional reconstitution for the SacII(-)/Aug(-) vector should be 2.9-fold greater than for the RI(-)/Aug(-) vector (reverse transcription of 810 nt or 276 nt without RT switch needed for reconstitution of RI(-)/Aug(-) or SacII(-)/Aug(-), respectively -- assuming equal recombination rates).
2) Lines200-203: Please mention that similar rates of RRE deletion are expected for both vectors, because both contain the same repeat sequences (with the exception of the mutated restriction sites) and the same interjacent sequence.
3) Lines 201-203 and/or Lines 234-236: Please discuss that 80 % RRE deletion means that 20 % of the transduction events happens with an RRE-positive vector. Therefore, this strategy should be optimized further (stated in Line 236) and would be best suited as an additional safety measure in SIN vectors.

---

## Round 0.2 · accepted · Accept

No further changes are required